# Dual Swap Disentangling

**Zunlei Feng**
Zhejiang University
zunleifeng@zju.edu.cn

**Xinchao Wang**
Stevens Institute of Technology
xinchao.wang@stevens.edu

**Chenglong Ke**
Zhejiang University
chenglongke@zju.edu.cn

**Anxiang Zeng**
Alibaba Group
renzhong@taobao.com

**Dacheng Tao**
University of Sydney
dctao@sydney.edu.au

**Mingli Song**[*]
Zhejiang University
brooksong@zju.edu.cn

## Abstract

Learning interpretable disentangled representations is a crucial yet challenging task. In this paper, we propose a weakly semi-supervised method, termed as *Dual Swap Disentangling (DSD)*, for disentangling using both labeled and unlabeled data. Unlike conventional weakly supervised methods that rely on full annotations on the group of samples, we require only limited annotations on paired samples that indicate their shared attribute like the color. Our model takes the form of a dual autoencoder structure. To achieve disentangling using the labeled pairs, we follow a "encoding-swap-decoding" process, where we first swap the parts of their encodings corresponding to the shared attribute, and then decode the obtained hybrid codes to reconstruct the original input pairs. For unlabeled pairs, we follow the "encoding-swap-decoding" process twice on designated encoding parts and enforce the final outputs to approximate the input pairs. By isolating parts of the encoding and swapping them back and forth, we impose the dimension-wise modularity and portability of the encodings of the unlabeled samples, which implicitly encourages disentangling under the guidance of labeled pairs. This dual swap mechanism, tailored for semi-supervised setting, turns out to be very effective. Experiments on image datasets from a wide domain show that our model yields state-of-the-art disentangling performances.

## 1 Introduction

Disentangling aims at learning dimension-wise interpretable representations from data. For example, given an image dataset of human faces, disentangling should produce representations or encodings for which part corresponds to interpretable attributes like facial expression, hairstyle, and color of the eye. It is therefore a vital step for many machine learning tasks including transfer learning (Lake et al. [2017]), reinforcement learning (Higgins et al. [2017a]) and visual concepts learning (Higgins et al. [2017b]).

Existing disentangling methods can be broadly classified into two categories, supervised approaches and unsupervised ones. Methods in the former category focus on utilizing annotated data to explicitly supervise the input-to-attribute mapping. Such supervision may take the form of partitioning the data into subsets which vary only along some particular dimension (Kulkarni et al. [2015], Bouchacourt et al. [2017]), or labeling explicitly specific sources of variation of the data (Kingma et al. [2014], Siddharth et al. [2017], Perarnau et al. [2016], Wang et al. [2017]). Despite their promising results, supervised methods, especially for deep-learning ones, usually require a large number of training samples which are often expensive to obtain.

---

[*]Corresponding author.

Unsupervised methods, on the other hand, do not require annotations but yield disentangled representations that are usually uninterpretable and dimension-wise uncontrollable. In other words, the user has no control over the semantic encoded in each dimension of the obtained codes. Taking a mugshot for example, the unsupervised approach fails to make sure that one of the disentangled codes will contain the feature of eye color. In addition, existing methods produce for each attribute with a single-dimension code, which sometimes has difficulty in expressing intricate semantics.

In this paper, we propose a *weakly semi-supervised* learning approach, dubbed as *Dual Swap Disentangling (DSD)*, for disentangling that combines the best of the two worlds. The proposed DSD takes advantage of limited annotated sample pairs together with many unannotated ones to derive dimension-wise and semantic-controllable disentangling. We implement the DSD model using an autoencoder, training on both labeled and unlabeled input data pairs and by swapping designated parts of the encodings. Specifically, DSD differs from the prior disentangling models in the following aspects.

- Limited Weakly-labeled Input Pairs. Unlike existing supervised and semi-supervised models that either require strong labels on each attribute of each training sample (Kingma et al. [2014], Perarnau et al. [2016], Siddharth et al. [2017], Wang et al. [2017], Banijamali et al. [2017]), or require fully weak labels on a group of samples sharing the same attribute (Bouchacourt et al. [2017]), our model only requires *limited pairs of samples*, which are much cheaper to obtain.

- Dual-stage Architecture. To our best knowledge, we propose the first dual-stage network architecture to utilize unlabeled sample pairs for semi-supervised disentangling, to facilitate and improve over the supervised learning using a small number of labeled pairs.

- Multi-dimension Attribute Encoding. We allow multi-dimensional encoding for each attribute to improve the expressiveness capability. Moreover, unlike prior methods (Kulkarni et al. [2015], Chen et al. [2016], Higgins et al. [2016], Burgess et al. [2017], Bouchacourt et al. [2017], Chen et al. [2018], Gao et al. [2018], Kim and Mnih [2018]), we do not impose any over-constrained assumption, such as each dimension being independent, into our encodings.

We show the architecture of DSD in Fig. 1. It comprises two stages, primary-stage and dual-stage, both are utilizing the same autoencoder. During training, the annotated pairs go through the primary-stage only, while the unannotated ones go through both. For annotated pairs, again, we only require weak labels to indicate which attribute the two input samples are sharing. We feed such annotated pairs to the encoder and obtained a pair of codes. We then designate which dimensions correspond to the specific shared attribute, and swap these parts of the two codes to obtain a pair of hybrid codes. Next we feed the hybrid codes to the decoder to reconstruct the final output of the labeled pairs. We enforce the reconstruction to approximate the input since we swap only the shared attribute, in which way we encourage the disentangling of the specific attribute in the designated dimensions and thus make our encodings dimension-wise controllable.

The unlabeled pairs during training go through both the primary-stage and the dual-stage. In the primary-stage, unlabeled pairs undergo the exact same procedure as the labeled ones, i.e., the encoding-swap-decoding steps. In the dual-stage, the decoded unlabeled pairs are again fed into the same autoencoder and parsed through the encoding-swap-decoding process for the second time. In other words, the code parts that are swapped during the primary-stage are swapped back in the second stage. With the guidance and constraint of labeled pairs, the dual swap strategy can generate informative feedback signals to train the DSD for the dimension-wise and semantic-controllable disentangling. The dual swap strategy, tailored for unlabeled pairs, turns out to be very effective in facilitating supervised learning with a limited number of samples.

Our contribution is therefore the first dual-stage strategy for semi-supervised disentangling. Also, require limited weaker annotations as compared to previous methods, and extend the single-dimension attribute encoding to multi-dimension ones. We evaluate the proposed DSD on a wide domain of image datasets, in term of both qualitative visualization and quantitative measures. Our method achieves results superior to the current state-of-the-art.

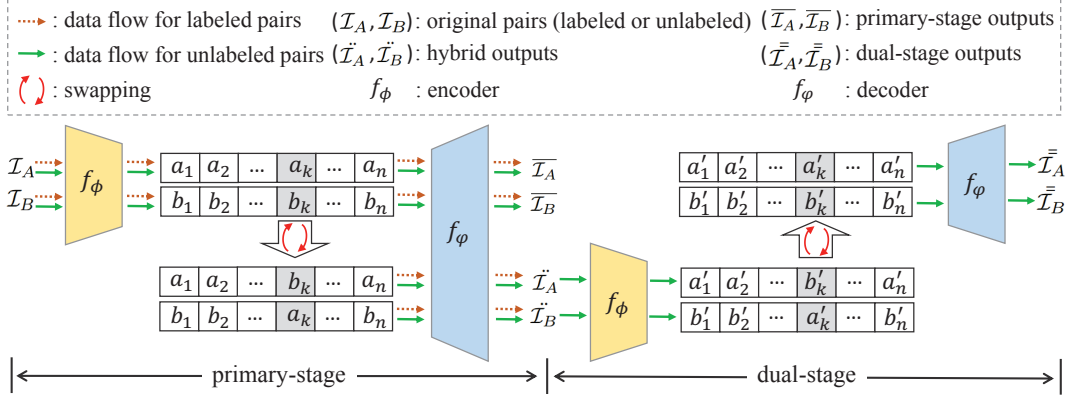

Figure 1: Architecture of the proposed DSD. It comprises two stages: primary-stage and dual-stage. The former one is employed for both labeled and unlabeled pairs while the latter is for unlabeled only.

## 2 Related Work

Recent works in learning disentangled representations have broadly followed two approaches, (semi-)supervised and unsupervised. Most of existing unsupervised methods (Burgess et al. [2017], Chen et al. [2018], Gao et al. [2018], Kim and Mnih [2018], Dupont [2018]) are based on the most two prominent methods InfoGAN (Chen et al. [2016]) and $\beta$-VAE (Higgins et al. [2016]). They however impose the independent assumption of the different dimensions of the latent code to achieve disentangling. Some semi-supervised methods (Bouchacourt et al. [2017], Siddharth et al. [2017]) import annotation information into $\beta$-VAE to achieve controllable disentangling. Supervised or semi-supervised methods like (Kingma et al. [2014], Perarnau et al. [2016], Wang et al. [2017], Banijamali et al. [2017], Feng et al. [2018]), they focus on utilizing annotated data to explicitly supervise the input-to-attribute mapping. Different with above methods, our method does not impose any over-constrained assumption and only require limited weak annotations.

We also give a brief review here about *swapping scheme*, *group labels*, and *dual mechanism*, which relate to our dual-stage model and weakly-labeled input. For *swapping*, Xiao et al. [2017] propose a supervised algorithm called DNA-GAN which can learn disentangled representations from multiple semantic images with swapping policy. The significant difference between our DSD and DNA-GAN is that the swapped codes correspond to different semantics in DNA-GAN. DNA-GAN requires lots of annotated multi-labeled images and the annihilating operation adopted by DNA-GAN is destructive. Besides, DNA-GAN is based on GAN, which also suffers from the unstable training of GAN. For *group information*, Bouchacourt et al. [2017] propose the Multi-Level VAE (ML-VAE) model for learning a meaningful disentanglement from a set of grouped observations. The group used in the ML-VAE requires that observations in the same group have the same semantics. However, it also has the limitation on increased reconstruction error. For *dual mechanism*, Zhu et al. [2017] use cycle-consistent adversarial networks to realize unpaired image-to-image translation. Xia et al. [2016] adopt the dual-learning framework for machine translation. However, they all require two domain entities, such as image domains (sketch and photo) and language domains (English and French). Different with above two works, our dual framework only needs one domain entity.

## 3 Method

In this section, we give more details of our proposed DSD model. We start by introducing the architecture and basic elements of our model, then show our training strategy for labeled and unlabeled pairs, and finally summarize the complete algorithm.

### 3.1 Dual-stage Autoencoder

The goal of our proposed DSD model is to take both weakly labeled and unlabeled sample pairs as input, and train an autoencoder that accomplishes dimension-wise controllable disentangling. We show a visual illustration of our model in Fig. 1, where the dual-stage architecture is tailored for the self-supervision on the unlabeled samples. In what follows, we describe DSD's basic elements: input, autoencoder, swap strategy and the dual-stage design in detail.

**Input** DSD takes a pair of samples as input denoted as $(\mathcal{I}_A, \mathcal{I}_B)$, where the pair can be either weakly labeled or unlabeled. Unlike conventional weakly supervised methods like Bouchacourt et al. [2017] that rely on full annotations on the group of samples, our model only requires limited and weak annotations as we only require the labels to indicate which attribute, if any, is shared by a pair of samples.

**Autoencoder** DSD conducts disentangling using an autoencoder trained in both stages. Given a pair of input $(\mathcal{I}_A, \mathcal{I}_B)$, weakly labeled or not, the encoder $f_\phi$ first encodes them to two vector representations $\mathcal{R}_A = f_\phi(\mathcal{I}_A) = [a_1, a_2, ..., a_n]$ and $\mathcal{R}_B = f_\phi(\mathcal{I}_B) = [b_1, b_2, ..., b_n]$, and then the decoder $f_\varphi$ decodes the obtained codes or encodings to reconstruct the original input pairs, i.e., $\overline{\mathcal{I}_A} = f_\varphi(\mathcal{R}_A)$ and $\overline{\mathcal{I}_B} = f_\varphi(\mathcal{R}_B)$. We would expect the obtained codes $\mathcal{R}_A$ and $\mathcal{R}_B$ to possess the following two properties: i) they include as much as possible information of the original input $\mathcal{I}_A$ and $\mathcal{I}_B$, and ii) they are disentangled and element-wise interpretable. The first property, as any autoencoder, is achieved through minimizing the following original autoencoder loss:

$$\mathcal{L}_\mathbf{o}(\mathcal{I}_A, \mathcal{I}_B; \phi, \varphi) = ||\mathcal{I}_A - \overline{\mathcal{I}_A}||_2^2 + ||\mathcal{I}_B - \overline{\mathcal{I}_B}||_2^2. \tag{1}$$

The second property is further achieved via the swapping strategy and dual-stage design, described in what follows.

**Swap Strategy** If given the knowledge that the pair of input $\mathcal{I}_A$ and $\mathcal{I}_B$ are sharing an attribute, such as the color, we can designate a specific part of their encodings, like $a_k$ of $\mathcal{R}_A$ and $b_k$ of $\mathcal{R}_B$, to associate the attribute semantic with the designated part. Assume that $\mathcal{R}_A$ and $\mathcal{R}_B$ are disentangled, swapping their code parts corresponding to the shared attribute, $a_k$ and $b_k$, should not change their encoding or their hybrid reconstruction $\ddot{\mathcal{I}}_A$ and $\ddot{\mathcal{I}}_B$. Conversely, enforcing the reconstruction after swapping to approximate the original input should facilitate and encourage disentangling for the specific shared attribute. Notably, here we allow each part of the encodings to be multi-dimensions, i.e., $a_k, b_k \in R^m, m \geq 1$, so as to improve the expressiveness of the encodings.

**Dual-stage** For labeled pairs, we know what their shared attribute is and can thus swap the corresponding parts of the code. For unlabeled ones, however, we do not have such knowledge. To take advantage of the large volume of unlabeled pairs, we implement a dual-stage architecture that allows the unlabeled pairs to swap random designated parts of their codes to produce the reconstruction during the primary-stage and then swap back during the second stage. Through this process, we explicitly impose the element-wise modularity and portability of the encodings of the unlabeled samples, and implicitly encourage disentangling under the guidance of labeled pairs.

### 3.2 Labeled Pairs

For a pair of labeled input $(\mathcal{I}_A, \mathcal{I}_B)$ in group $\mathcal{G}_k$, meaning that they share the attribute corresponding to the $k$-th part of their encodings $\mathcal{R}_A$ and $\mathcal{R}_B$, we swap their $k$-th part and get a pair of hybrid codes $\ddot{\mathcal{R}}_A = [a_1, a_2, ..., b_k, ..., a_n]$ and $\ddot{\mathcal{R}}_B = [b_1, b_2, ..., a_k, ..., b_n]$. We then feed the hybrid code pair $\ddot{\mathcal{R}}_A$ and $\ddot{\mathcal{R}}_B$ to the decoder $f_\varphi$ to obtain the final representation $\ddot{\mathcal{I}}_A$ and $\ddot{\mathcal{I}}_B$. We enforce the reconstructions $\ddot{\mathcal{I}}_A$ and $\ddot{\mathcal{I}}_B$ to approximate $(\mathcal{I}_A, \mathcal{I}_B)$, and encourage disentangling of the $k$-th attribute. This is achieved by minimizing the swap loss

$$\mathcal{L}_\mathbf{s}(\mathcal{I}_A, \mathcal{I}_B; \phi, \varphi) = ||\mathcal{I}_A - \ddot{\mathcal{I}}_A||_2^2 + ||\mathcal{I}_B - \ddot{\mathcal{I}}_B||_2^2, \tag{2}$$

so that the $k$-th part of $\mathcal{R}_A$ and $\mathcal{R}_B$ will only contain the shared semantic.

We take the total loss $\mathcal{L}_\mathbf{p}$ for the labeled pairs to be the sum of the original autoencoder loss $\mathcal{L}_\mathbf{o}$ and swap loss $\mathcal{L}_\mathbf{s}(\mathcal{I}_A, \mathcal{I}_B; \phi, \varphi)$:

$$\mathcal{L}_\mathbf{p}(\mathcal{I}_A, \mathcal{I}_B; \phi, \varphi) = \mathcal{L}_\mathbf{o}(\mathcal{I}_A, \mathcal{I}_B; \phi, \varphi) + \alpha \mathcal{L}_\mathbf{s}(\mathcal{I}_A, \mathcal{I}_B; \phi, \varphi), \tag{3}$$

where $\alpha$ is a balance parameter, which decides the degree of disentanglement.

---

**Algorithm 1** The Dual Swap Disentangling (DSD) algorithm

---

**Input:** Paired observation groups $\{\mathcal{G}_k, k = 1, 2, .., n\}$, unannotated observation set $\mathbb{G}$.
1: Initialize $\phi^1$ and $\varphi^1$.
2: **for** $t = 1, 3, ..., T$ epochs **do**
3:     Random sample $k \in \{1, 2, ..., n\}$.
4:     Sample paired observation $(\mathcal{I}_A, \mathcal{I}_B)$ from group $\mathcal{G}_k$.
5:     Encode $\mathcal{I}_A$ and $\mathcal{I}_B$ into $\mathcal{R}_A$ and $\mathcal{R}_B$ with encoder $f_{\phi^t}$.
6:     Swap the $k$-th part of $\mathcal{R}_A$ and $\mathcal{R}_B$ and get two hybrid representations $\ddot{\mathcal{R}}_A$ and $\ddot{\mathcal{R}}_B$.
7:     Construct $\mathcal{R}_A$ and $\mathcal{R}_B$ into $\bar{\mathcal{I}}_A = f_{\varphi^t}(\mathcal{R}_A)$ and $\bar{\mathcal{I}}_B = f_{\varphi^t}(\mathcal{R}_B)$.
8:     Construct $\ddot{\mathcal{R}}_A$ and $\ddot{\mathcal{R}}_B$ into $\ddot{\mathcal{I}}_A = f_{\varphi^t}(\ddot{\mathcal{R}}_A)$ and $\ddot{\mathcal{I}}_B = f_{\varphi^t}(\ddot{\mathcal{R}}_B)$.
9:     Update $\phi^{t+1}, \varphi^{t+1} \leftarrow \phi^t, \varphi^t$ by ascending the gradient estimate of $\mathcal{L}_{\mathbf{p}}(\mathcal{I}_A, \mathcal{I}_B; \phi^t, \varphi^t)$.
10:    Sample unpaired observation $(\mathcal{I}_A, \mathcal{I}_B)$ from unannotated observation set $\mathbb{G}$.
11:    Encode $\mathcal{I}_A$ and $\mathcal{I}_B$ into $\mathcal{R}_A$ and $\mathcal{R}_B$ with encoder $f_{\phi^{t+1}}$.
12:    swap the $k$-th part of $\mathcal{R}_A$ and $\mathcal{R}_B$ and get two hybrid representations $\ddot{\mathcal{R}}_A$ and $\ddot{\mathcal{R}}_B$.
13:    Construct $\mathcal{R}_A$ and $\mathcal{R}_B$ into $\bar{\mathcal{I}}_A = f_{\varphi^{t+1}}(\mathcal{R}_A)$ and $\bar{\mathcal{I}}_B = f_{\varphi^{t+1}}(\mathcal{R}_B)$.
14:    Construct $\ddot{\mathcal{R}}_A$ and $\ddot{\mathcal{R}}_B$ into $\ddot{\mathcal{I}}_A = f_{\varphi^{t+1}}(\ddot{\mathcal{R}}_A)$ and $\ddot{\mathcal{I}}_B = f_{\varphi^{t+1}}(\ddot{\mathcal{R}}_B)$.
15:    Encode $(\ddot{\mathcal{I}}_A, \ddot{\mathcal{I}}_B)$ into $\ddot{\mathcal{R}}'_A$ and $\ddot{\mathcal{R}}'_B$ with encoder $f_{\phi^{t+1}}$.
16:    Swap the $k$-th parts of $\ddot{\mathcal{R}}'_A$ and $\ddot{\mathcal{R}}'_B$ backward and get $\mathcal{R}'_A$ and $\mathcal{R}'_B$.
17:    Construct $\mathcal{R}'_A$ and $\mathcal{R}'_B$ into $\bar{\bar{\mathcal{I}}}_A = f_{\varphi^{t+1}}(\mathcal{R}'_A)$ and $\bar{\bar{\mathcal{I}}}_B = f_{\varphi^{t+1}}(\mathcal{R}'_B)$.
18:    Update $\phi^{t+2}, \varphi^{t+2} \leftarrow \phi^{t+1}, \varphi^{t+1}$ by ascending the gradient estimate of $\mathcal{L}_{\mathbf{u}}(\mathcal{I}_A, \mathcal{I}_B; \phi^{t+1}, \varphi^{t+1})$.
19: **end for**
**Output:** $\phi^T, \varphi^T$

---

### 3.3 Unlabeled Pairs

Unlike the labeled pairs that go through only the primary-stage, unlabeled pairs go through both the primary-stage and the dual-stage, in other words, the "encoding-swap-decoding" process is conducted twice for disentangling. Like the labeled pairs, in the primary-stage the unlabeled pairs $(\mathcal{I}_A, \mathcal{I}_B)$ also produce a pair of hybrid outputs $\ddot{\mathcal{I}}_A$ and $\ddot{\mathcal{I}}_B$ through swapping a random $k$-th part of $\mathcal{R}_A$ and $\mathcal{R}_B$. In the dual-stage, the two hybrids $\ddot{\mathcal{I}}_A$ and $\ddot{\mathcal{I}}_B$ are again fed to the same encoder $f_\phi$ and encoded as new representations $\ddot{\mathcal{R}}'_A = [a'_1, a'_2, ..., b'_k, ..., a'_n]$ and $\ddot{\mathcal{R}}'_B = [b'_1, b'_2, ..., a'_k, ..., b'_n]$. We then swap back the $k$-th part of $\ddot{\mathcal{R}}'_A$ and $\ddot{\mathcal{R}}'_B$ and denote the new codes as $\mathcal{R}'_A = [a'_1, a'_2, ..., a'_k, ..., a'_n]$ and $\mathcal{R}'_B = [b'_1, b'_2, ..., b'_k, ..., b'_n]$. These codes are fed to the decoder $f_\varphi$ to produce the final output $\bar{\bar{\mathcal{I}}}_A = f_\varphi(\mathcal{R}'_A)$ and $\bar{\bar{\mathcal{I}}}_B = f_\varphi(\mathcal{R}'_B)$.

We minimize the reconstruction error of dual swap output with respect to the original input, and write the dual swap loss $\mathcal{L}_{\mathbf{d}}$ as follows:

$$\mathcal{L}_{\mathbf{d}}(\mathcal{I}_A, \mathcal{I}_B; \phi, \varphi) = ||\mathcal{I}_A - \bar{\bar{\mathcal{I}}}_A||_2^2 + ||\mathcal{I}_B - \bar{\bar{\mathcal{I}}}_B||_2^2. \tag{4}$$

The dual swap reconstruction minimization here provides a unique form of self-supervision. That is, by swapping random parts back and forth, we encourage the element-wise separability and modularity of the obtained encodings, which further helps the encoder to learn disentangled representations under the guidance of limited weak labels.

The total loss for the unlabeled pairs consists of the original autoencoder loss $\mathcal{L}_{\mathbf{o}}(\mathcal{I}_A, \mathcal{I}_B; \phi, \varphi)$ and dual autoencoder loss $\mathcal{L}_{\mathbf{d}}(\mathcal{I}_A, \mathcal{I}_B; \phi, \varphi)$:

$$\mathcal{L}_{\mathbf{u}}(\mathcal{I}_A, \mathcal{I}_B; \phi, \varphi) = \mathcal{L}_{\mathbf{o}}(\mathcal{I}_A, \mathcal{I}_B; \phi, \varphi) + \beta \mathcal{L}_{\mathbf{d}}(\mathcal{I}_A, \mathcal{I}_B; \phi, \varphi), \tag{5}$$

where $\beta$ is the balance parameter. As we will show in our experiment, adopting the dual swap on unlabeled samples and solving the objective function of Eq. 5, yield a significantly better result as compared to only using unlabeled samples during the primary-stage without swapping, which corresponds to optimizing over the autoencoder loss alone.

### 3.4 Complete Algorithm

Within each epoch during training, we alternatively optimize the autoencoder using randomly-sampled labeled and unlabeled pairs. The complete algorithm is summarized in Algorithm 1. Once trained, the encoder is able to infer disentangled encodings that can be applied in many applications.

## 4 Experiments

To validate the effectiveness of our methods, we conduct experiments on six image datasets of different domains: a synthesized Square dataset, Teapot (Moreno et al. [2016], Eastwood and Williams [2018]), MNIST (Haykin and Kosko [2009]), dSprites (Higgins et al. [2016]), Mugshot (Shen et al. [2016]), and CAS-PEAL-R1 (Gao et al. [2008]). We firstly qualitatively assess the visualization of DSD's generative capacity by performing swapping operation on the parts of latent codes, which verifies the *disentanglement* and *completeness* of our method. To evaluate the *informativeness* of the disentangled codes, we compute the classification accuracies based on DSD encodings. We are not able to use the framework of Eastwood and Williams [2018] as it is only applicable to methods that encode each semantic into a single dimension code. In the DSD framework, the latent code's length and semantic number for the six datasets are set as follows: Square $(15, 3)$, Teapot $(50, 5)$, MNIST $(15, 3)$, dSprites $(25, 5)$, CAS-PEAL-R1 $(40, 4)$ and Mugshot $(100, 2)$. The latent code's length is empirically set, but usually set larger for sophisticated attributes.

In our experiment, the visual results are generated with the $64 \times 64$ network architecture and other quantitative results are generated with the $32 \times 32$ network architecture. For the $32 \times 32$ network architecture, the encoder / discriminatior ($\mathcal{D}$) / auxiliary network ($\mathcal{Q}$) and the decoder / generator ($\mathcal{G}$) are shown in Table 1. The $64 \times 64$ network architecture is same as architecture of Eastwood and Williams [2018]. Adam optimizer (Kingma and Ba [2014]) is adopted with learning rates of $1e^{-4}$ ($64 \times 64$ network) and $0.5e^{-4}$ ($32 \times 32$ network). The batch size is 64. For the stable training of InfoGAN, we fix the latent codes' standard deviations to 1 and use the objective of the improved Wasserstein GAN (Gulrajani et al. [2017]), simply appending InfoGAN's approximate mutual information penalty. We use layer normalization instead of batch normalization. For the above two network architecture, $\alpha$ and $\beta$ are all set as 5 and 0.2, respectively.

| Encoder / $\mathcal{D}$/$\mathcal{Q}$ | Decoder /$\mathcal{G}$ |
|---|---|
| $3 \times 3$ 32 conv. | FC $4 \cdot 4 \cdot 8 \cdot 32$ |
| BN, ReLU, $3 \times 3$ 32 conv<br>BN, ReLU, $3 \times 3$ 64 conv, $\downarrow$ | BN, ReLU, $3 \times 3$ 256 conv, $\uparrow$<br>BN, ReLU, $3 \times 3$ 128 conv |
| BN, ReLU, $3 \times 3$ 64 conv<br>BN, ReLU, $3 \times 3$ 128 conv, $\downarrow$ | BN, ReLU, $3 \times 3$ 128 conv, $\uparrow$<br>BN, ReLU, $3 \times 3$ 64 conv |
| BN, ReLU, $3 \times 3$ 128 conv<br>BN, ReLU, $3 \times 3$ 256 conv, $\downarrow$ | BN, ReLU, $3 \times 3$ 64 conv, $\uparrow$<br>BN, ReLU, $3 \times 3$ 32 conv |
| FC Output | BN, ReLU, $3 \times 3$ 3 conv, tanh |

Table 1: Network architecture for image size $32 \times 32$. Each network has 3 residual blocks (all but the first and last rows). The input to each residual block is added to its output (with appropriate downsampling/upsampling to ensure that the dimensions match). Downsampling $\downarrow$ is performed with mean pooling and $\uparrow$ indicates nearest-neighbour upsampling.

### 4.1 Qualitative Evaluation

We show in Fig. 2 some visualization results on the six datasets. For each dataset, we show input pairs, the swapped attribute, and results after swapping.

**Square** We create a synthetic image dataset of $60,000$ image samples ($30,000$ pair images), where each image features a randomly-colored square at a random position with a randomly-colored background. The training, validation and testing dataset are set as $\{(20,000), (9,000)$ and $(1,000)\}$, respectively. Visual results of DSD on Square dataset are shown in Fig. 2(a), where DSD leads to visually plausible results.

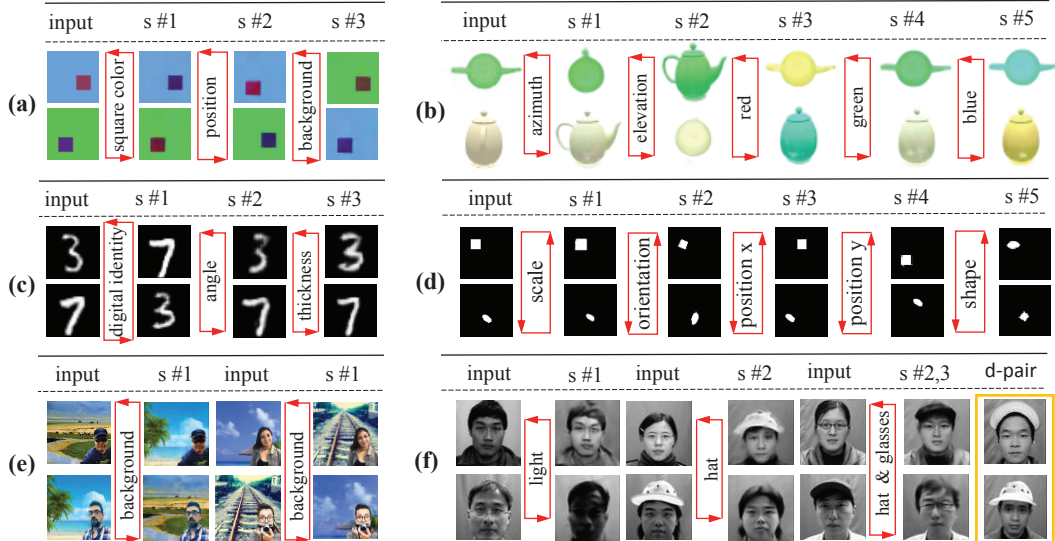

Figure 2: Visual results on six datasets: (a) Square, (b) Teapot, (c) MNIST, (d) dSprites, (e) Mugshot, and (f) CAS-PEAL-R1. "d-pair" indicates disturbed pair.

**Teapot** The Teapot dataset used in Eastwood and Williams [2018] contains $200,000$ $64 \times 64$ color images of a teapot with varying poses and colors. Each generative factor is independently sampled from its respective uniform distribution: azimuth $(z_0) \sim U\lceil 0, 2\pi \rceil$, elevation $(z_1) \sim U\lceil 0, 2\pi \rceil$, red $(z_2) \sim U\lceil 0; 1 \rceil$, green $(z4) \sim U\lceil 0; 1 \rceil$. In the experiment, we used $50,000$ training, $10,000$ validation and $10,000$ testing samples. Fig. 2(b) shows the visual results on Teapot, where we can see that the five factors are evidently disentangled.

**MNIST** In the visual experiment, we adopt InfoGAN to generate $5,000$ paired samples, for which we vary the following factors: digital identity $(0-9)$, angle and stroke thickness. The whole training dataset contains $50,000$ samples: $5,000$ generated paired samples and $45,000$ real unpaired samples collected from the original dataset. Semantics swapping for MNIST are shown in Fig. 2(c), where the digits swap one attribute but preserve the other two. For example, when swapping the angle, the digital identity and thickness are kept unchanged. The generated images again look very realistic.

**dSprites** The dSprites is a dataset of 2D shapes procedurally generated from 6 ground truth independent latent factors. These factors are color (white), shape (heart, oval and square), scale (6 values), rotation (40 values), position X (32 values) and position Y (32 values) of a sprite. All possible combinations of these factors are present exactly once, generating N = 737280 total images. We sample $100,000$ pairs from original dSprites, which are divided into $\{(80,000), (10,000), (10,000)\}$ for training, validation and testing. Fig. 2(d) shows the visual results with swapped above latent factors, where we can see that the five factors are again obviously disentangled.

**Mugshot** We also use the Mugshot dataset which contains selfie images of different subjects with different backgrounds. This dataset is generated by artificially combining human face images in Shen et al. [2016] with $1,000$ scene photos collected from internet. For Mugshot dataset, we divided it into $\{(20,000), (9,000), (1,000)\}$ for training, validation and testing. Fig. 2(e) shows the results of the same mugshot through swapping different backgrounds, which are visually impressive. Note that, in this case we only consider two semantics, the foreground being the human selfie and the background being the collected scene. The good visual results can be partially explained by the fact that the background with different subjects has been observed by DSD during training.

**CAS-PEAL-R1** CAS-PEAL-R1 contains $30,900$ images of $1,040$ subjects, of which $438$ subjects wear 6 different types of accessories (3 types of glasses, and 3 types of hat). There are images of 233 subjects that involve at least 10 lighting changes and at most 31 lighting changes. We sample $50,000$ pair samples from original CAS-PEAL-R1. They are divided into $\{(40,000), (9,000), (1,000)\}$ for training, validation and testing. Fig. 2(f) shows the visual results with swapped light, hat and glasses.

Notably, the covered hair by the hats can also be reconstructed when the hats are swapped, despite the qualities of hybrid images are exceptional. This can be in part explained by the existence of disturbed paired samples, as depicted in the last column. This pair of images is in fact labeled as sharing the same hat, although the appearances of the hats such as the wearing angles are significantly different, making the supervision very noisy.

## 4.2 Quantitative Evaluation

To quantitatively evaluate the *informativeness* of disentangled codes, we compare our methods with 4 methods: InfoGAN (Chen et al. [2016]), $\beta$-VAE (Higgins et al. [2016]), Semi-VAE (Siddharth et al. [2017]) and basic Autoencoder. We first use InfoGAN to generate $5,0000$ pair digital samples, and then train all methods on this generated dataset. For InfoGAN and $\beta$-VAE , the lengths of their codes are set as 5. To fairly compare with the above two methods, the codes' length of Semi-VAE, Autoencoder and our DSD are taken to be $5 \times 3$, which means the code contains 3 parts and each part's length is 5. In this condition, we can compare part of codes ($length = 5$) that correspond to digit identity with whole codes ($length = 5$) of InfoGAN and $\beta$-VAE and variable ($length = 1$) that correspond to digit identity. For the basic Autoencoder, the highest accuracy part is treated as the identity part. After training all the models, real MNIST data are encoded as codes. Then, $55,000$ training samples are used to train a simple knn classifier and remaining $10,000$ are used as test samples. Table 2 gives the classification accuracy of different methods, where the Info-GAN achieves the worst accuracy score. The DSD(0.5) achieves best accuracy score, which further validates the informativeness of our DSD.

Table 2: The accuracy score comparison among different models. DSD(n) denotes the DSD with $n$ supervision rate paired samples. Accuracy (ACC) values are shown as "$q/p$", where $q$ is the accuracy obtained using the digital identity part of the codes for classification, and $p$ is the accuracy obtained using the whole codes.

| Model | $\beta$-VAE($\beta$=1) | $\beta$-VAE($\beta$=6) | InfoGAN | Semi-VAE | Autoencoder | DSD(0.5) | DSD(1) |
|---|---|---|---|---|---|---|---|
| ACC | 0.22/0.72 | 0.25/0.71 | 0.19/0.51 | 0.22/0.57 | 0.66/0.93 | **0.76**/0.91 | 0.742/0.90 |

In addition, to compare the annotated dataset's requirement of different (semi-)supervised methods, we summarize it in Table 3. Name abbreviation with corresponding methods is given as following: DC-IGN (Kulkarni et al. [2015]), DNA-GAN (Xiao et al. [2017]), TD-GAN (Wang et al. [2017]), Semi-DGM (Kingma et al. [2014]), Semi-VAE (Siddharth et al. [2017]), ML-VAE (Bouchacourt et al. [2017]), JADE (Banijamali et al. [2017]). DSD is the only one that requires limited and weak labels, meaning that it requires the least amount of human annotations.

Table 3: Comparison of the required annotated data. **Label** indicates whether the method require strong label or weak label. **Rate** indicates the proportion of annotated data required for training.

| | DC-IGN | DNA-GAN | TD-GAN | Semi-DGM | Semi-VAE | JADE | ML-VAE | DSD |
|---|---|---|---|---|---|---|---|---|
| **Label** | strong | strong | strong | strong | strong | strong | *weak* | *weak* |
| **Rate** | 100 % | 100 % | 100 % | *limited* | *limited* | *limited* | 100 % | *limited* |

## 4.3 Supervision Rate

We also conduct experiments to demonstrate the impact of the supervision rate for DSD's disentangling capabilities, where we set the rates to be $0.0, 0.1, 0.2, ..., 1.0$. From Fig. 3(a), we can see that different supervision rates do not affect the convergence of DSD. Lower supervision rate will however lead to the overfitting if the epoch number greater than the optimal one. Fig. 3(d) shows the classification accuracy of DSD with different supervision rates. With only $20\%$ paired samples, DSD achieves comparable accuracy as the one obtained using $100\%$ paired data, which shows that the dual-learning mechanism is able to take good advantage of unpaired samples. Fig. 3(c) shows some hybrid images that are swapped the digital identity code parts. Note that, images obtained by DSD with supervision rates equal to $0.2, 0.3, 0.4, 0.5$ and $0.7$ keep the angles of the digits correct while others not. These image pairs are highlighted in yellow.

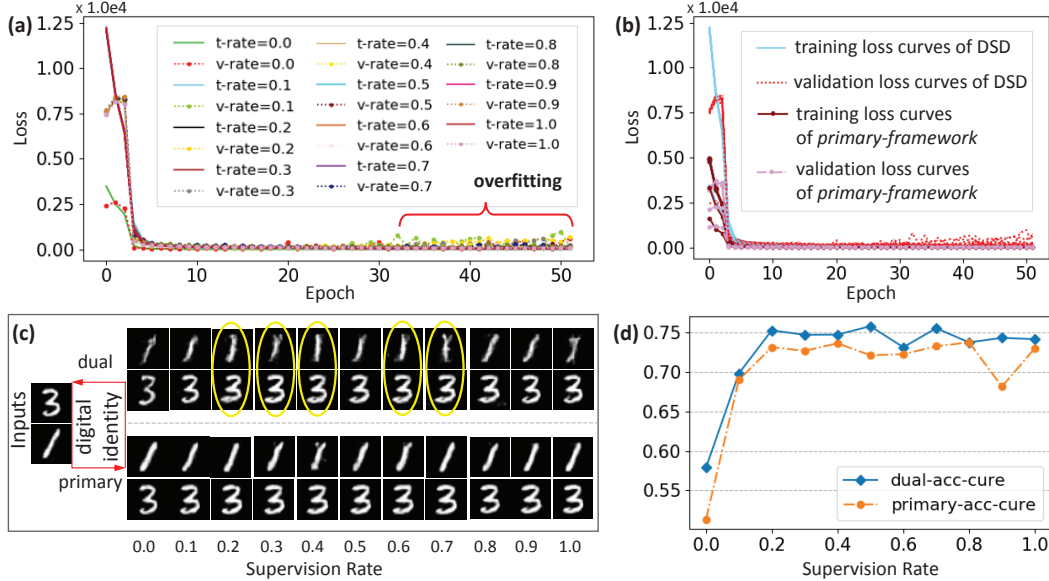

Figure 3: Results of different supervision rate. (a) The training loss curves and validation loss curves of different supervision rates, where "t-rate" indicates training loss of supervision rate and "v-rate" indicates validation loss of supervision rate. (b) The training and validation loss curves of the DSD (dual framework) and *primary-framework* with different supervision rates. (c) Visual results of different supervision rates through swapping parts of codes that correspond to the digital identities. (d) Classification accuracy of codes that are encoded by DSD with different supervision rate.

## 4.4 Primary vs Dual

To verify the effectiveness of dual-learning mechanism, we compare our DSD (dual framework) with a basic *primary-framework* that only contains primary-stage. The *primary-framework* also requires paired and unpaired samples. The major difference between the *primary-framework* and DSD is that there is no swapping operation for unpaired samples in the *primary-framework*. Fig. 3(b) gives the training and validation loss curves of the DSD and *primary-framework* with different supervision rates, where we can find that different supervision rates have no visible impacts on the convergence of DSD and *primary-framework*. From Fig. 3(d), we can see that accuracy scores of the DSD are always higher than accuracies of the *primary-framework* in different supervision rate, which proves that codes disentangled by the DSD are more informative than those disentangled by the *primary-framework*. Fig. 3(c) gives the visual comparison between the hybrid images in different supervision rate. It is obvious that hybrid images of the *primary-framework* are almost the same with original images, which indicates that the swapped codes contain redundant angle information. In other words, the disentanglement of the *primary-framework* is defective. On the contrary, most of the hybrid images of DSD keep the angle effectively, indicating that swapped coded only contains the digital identity information. These results show that the DSD is indeed superior to the *primary-framework*.

## 5 Discussion and Conclusion

In this paper, we propose the Dual Swap Disentangling (DSD) model that learns disentangled representations using limited and weakly-labeled training samples. Our model requires the shared attribute as the only annotation of a pair of input samples, and is able to take advantage of the vast amount of unlabeled samples to facilitate the model training. This is achieved by the dual-stage architecture, where the labeled samples go through the "encoding-swap-decoding" process once while the unlabeled ones go through the process twice. Such self-supervision mechanism for unlabeled samples turns out to be very effective: DSD yields results superior to the state-of-the-art on several datasets of different domains. In the future work, we will take semantic hierarchy into consideration and potentially learn disentangled representations with even fewer labeled pairs.

**Acknowledgments**

This work is supported by Natonal Basic Research Program of China under Grant No. 2015CB352400, National Natural Science Foundation of China (61572428,U1509206), Fundamental Research Funds for the Central Universities (2017FZA5014), Key Research and Development Program of Zhejiang Province (2018C01004), and Australian Research Council Projects (FL-170100117, DP-140102164).

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
