[Supplementary Material · NeurIPS_2018_suplement.pdf]

# 1 Quantitative Evaluation on dSprites

In the additional experiments, we adopt the enhanced disentanglement metric (Kim and Mnih [2018]) to quantitatively compare the disentangling performance of DSD against other methods:$\beta$-VAE (Higgins et al. [2016]), InfoGAN (Chen et al. [2016]), DC-IGN (Kulkarni et al. [2015]), FactorVAE (Kim and Mnih [2018]). The enhanced disentanglement metric is based on the metric proposed by Higgins et al. [2016]. The procedure of the enhanced disentanglement metric is as follows: Choose a factor k; generate data with this factor fixed but all other factors varying randomly; obtain their representations; rescale each dimension by its empirical standard deviation of representations over the full data (or a large enough random subset); take the empirical variance in each dimension. Then the index of the dimension with lowest variance gives one training input with training output k for a classifier. So if the representation is perfectly disentangled, the empirical variance in the dimension corresponding to the fixed factor will be 0. Table 1 gives the metric score of different methods, where the FactorVAE($\gamma$=50) achieves the best accuracy score and Our DSD(0.5) achieves the second-best accuracy score. What's more, the DSD with $50\%$ supervision rate achieves higher accuracy score than DSD with $100\%$ supervision rate, which verifies that the semi-supervised DSD has better robustness.

Table 1: Disentanglement metric classification accuracy for dSprites dataset. 'EDMS' indicates the Enhanced Disentanglement Metric Score (Kim and Mnih [2018]. DSD(n) denotes the DSD with $n$ supervision rate paired samples).

| Model | $\beta$-VAE($\beta$=1) | $\beta$-VAE($\beta$=6) | InfoGAN | DC-IGN | FactorVAE($\gamma$=50) | DSD(0.5) | DSD(1) |
|---|---|---|---|---|---|---|---|
| **EDMS** | 0.6713 | 0.7894 | 0.6152 | 0.9285 | **0.9436** | 0.9317 | 0.9153 |

# 2 Additional Qualitative Results

To further verify the effectiveness of our DSD, more visualization results are given in the following Figures. For each dataset, we show the input pairs, the swapped attribute, the results after swapping, reconstructed results of primary-stage, and reconstructed results of dual-stage, respectively.

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

Figure 1: Visual results on Square. Name abbreviation with corresponding semantics is given as following: 'ssc.' (small square color), 'po.' (small square position), 'bk.' (background color), 'primary' (primary-stage output), 'dual' (dual-stage output).

Figure 2: Visual results on MNIST. 'id' indicates digital identity, 'thic.' indicates thickness, 'prim.' means the reconstructed result of primary-stage.

Figure 3: Visual results on Teapot.

| input | hybrid | primary | dual | input | hybrid | primary | dual |

Figure 4: Visual results on CAS-PEAL-R1.

| input1 | i #1 | i #2 | i #3 | i #4 | i #5 | i #6 | i #7 | i #8 |
|--------|------|------|------|------|------|------|------|------|

Figure 5: Visual results on Mugshot.