[Reviews · NeurIPS 2018]

Reviewer 1



This paper describes a novel algorithm for learning disentangled representations in an autoencoder architecture. The methods relies on the availability of at least 10% of weakly supervised pairs of images (where for each pair the common factor is labelled). The method then uses this information to perform a swap in the latent encoding, where the latents swapped are assigned to represent the common generative factor. This encourages disentanglement and bootstraps the second "dual-stage" part of the approach, which performs double swaps on pairs of images that do not contain any labelled information. The approach appears to perform well on a number of datasets, however the paper requires further quantitative evaluation against the state of the art baselines in order to be fully convincing. Suggestions for improvement: -- The authors are encouraged to cite DC-IGN by Kulkarni et al (2015) as another weakly-labelled approach to disentangled representation learning (Lines 48-53 and 239). It would also be useful to run this approach as a baseline. -- Line 146. The authors mention using m>=1 latent dimensions to represent each of the generative factors. The paper would benefit from a discussion on how m is set. Is m fixed to be the same for all generative factors? How does the performance of the approach change with various values of m? How do the authors select which subset of the latent representation corresponds to each generative factor? In particular, how do the authors make sure that the random swaps in the dual stage do not cross the factor boundaries? Are all latent dimensions allocated to represent certain generative factors or are certain dimensions set to be "free"? What happens to these dimensions in the dual stage? -- How many supervised examples were seen in the CAS-PEAL-R1 and Mugshot experiments? -- When training the beta-VAE baseline, the authors used 5 latent dimensions. This is too low for achieving a good performance with a beta-VAE. The authors are encouraged to re-run the experiments with at least 10 latent dimensions for MNIST. We have run similar experiments with a beta-VAE and found that we could get up to 90% accuracy on MNIST classification using the latent space of a pre-trained disentangled beta-VAE. -- Line 245. I am not sure what the authors meant by "variable (length = 1) that correspond to digit identity" -- How do the authors separate out the digit identity part of the encoding for the Autoencoder baseline? -- Figure 3. The plot is not particularly useful, since it is hard to spot the difference between the different versions of the model. I suggest that the authors move the figure to the appendix and replace it with a table of quantitative results comparing the disentanglement score (e.g. from Kim and Mnih, 2018) of DSD and beta-VAE trained on the dSprites dataset (Matthey et al, 2017). The dSprites dataset is a unit test of disentanglement with clearly established metrics. It is also a dataset that comes with the ground truth labels for all the factors of variation, and where beta-VAE struggles to learn the sprite identity information. If the authors are able to demonstrate that their approach achieves better disentanglement scores than beta-VAE on this dataset while also learning about object identity at various levels of supervision, it will be a very powerful and persuasive demonstration of the DSD performance. I am willing to increase my score if this demonstration is provided. -- Section 4.4. I don't find this section particularly convincing, in particular given the strong wording ("proves") used by the authors. I would recommend running the quantitative analysis of the primary vs dual ablation as suggested above, preferably with error bounds to justify the strong conclusions of this section. Minor points: -- Lines 35-37. The authors mention that the unsupervised methods to disentanglement learn representations that are "uninterpretable", where the methods may fail to represent hair colour when learning about faces. The authors are advised to reword that paragraph, since the choice of words and example are mis-representative. For example, the beta-VAE approach (Higgins et al, 2017) learnt to represent hair colour on the CelebA dataset. Furthermore, the approaches would not be published if they learnt "uninterpretable" representations, since this is the whole point of disentangled representation learning. -- Line 38. contains --> contain -- Line 66. "which attribute of the two input samples is sharing" --> "which attribute the two input samples are sharing" -- Line 110. unpair --> unpaired -- Line 153. encourages --> encourage -- Line 181. pairs is consists --> pairs consists -- Line 190. "is able to conduct disentangled encodings" --> "is able to infer (??) disentangled encodings" -- Line 212. of an teapot --> of a teapot -- Line 225. hairs --> hair -- Line 226. are not exceptional --> being not exceptional -- Table 1 caption. digital --> digit -- Line 289. disentangled with --> disentangled representations with -- Line 273. are informativeness --> are more informative ------------------------- After reading the authors' response, I have increased my mark to 7. I believe that the paper warrants an acceptance.

Reviewer 2



This paper addresses the problem of disentangled representation learning in a very specific way, namely with a focus on “interpretability” and “controllability” of the representations -- such that the meaning of the individual (group of) latents can be understood by a human (interpretability), and that the network designer can control what is, and what is not, to be contained in the disentangled representations. The chosen approach is to start with “readily available” ground truth attribute labels, then carve up the latent representation space in pre-assigned groups, one group per attribute to be disentangled. Then, to encourage the encoder to actually assign the corresponding attribute encoding only to the pre-selected group -- and not to any other parts of the representation space -- a generator is expected to still reconstruct the image perfectly when the latent representation of that attribute is replaced with that of another image with the same attribute value. This requires a sufficient number of pairs of images for which the attribute values are known and identical. In addition, for pairs of images for which the attribute values are not known (which can be the majority of the training data), a randomly chosen group is swapped between the encodings, then decoded and the result encoded again, followed by “swapping back” the same group, before again decoding. If the representation is disentangled, then this should results in the original images to be reconstructed after the second stage. Pros: The idea of swapping latents and using the outcome as a learning signal for disentangled representation learning is a very good one, and I have not seen it presented in publications before. The method clearly works well on the datasets presented, and with the ground-truth label information provided. The visual results of this approach (Figure 2) look very nice, and do reflect desired properties of disentangled representations, namely the compositionality resulting from the possibility to change one semantically meaningful part of the representation independently of all others. Cons: I have a number of concerns with this approach, or at least with the way it is portrayed. Some of those concerns can be highlighted by revisiting the claimed contributions of this work: - “Our contribution is therefore the first dual-stage strategy for semi-supervised disentangling.” -- It is not clear to me how this a contribution by itself. - “Also, require limited weaker annotations as compared to previous methods” -- this is obviously only true compared to previous supervised methods, despite the authors pointing out early on that the most prominent successful methods are actually unsupervised - “and extend the single-dimension attribute encoding to multi-dimension ones.” -- I can see nothing in this work that shows that this is actually an advantage per se - “Our method achieves results superior to the current state-of-the-art.” -- In order to talk about superior results, a fair comparison is required. Adding supervised information (even if “weakly supervised”) solves a rather different kind of problem than what fully unsupervised methods try to achieve. I assume this statement is meant to be corroborated by the numbers shown in Table 1, but it is unclear to me how these very different models (with very different amounts of information provided to the networks, with InfoGAN and beta-VAE being unsupervised) can be fairly compared to each other. On that note, I could not figure out from the text what beta-VAE(1) and beta-VAE(6) stand for. - Line 34: “Unsupervised methods, on the other hand, do not require annotations but yield disentangled representations that are usually uninterpretable and dimension-wise uncontrollable.” The statement that the representations of unsupervised methods (such as InfoGAN and beta-VAE) are “usually uninterpretable” is patently false, and that they are “uncontrollable” is meaningless, as the aim of unsupervised methods is the unguided discovery of disentangled factors, rather than forced assignments of known or presumed factors. - Line 41: “In this paper, we propose a weakly semi-supervised learning approach, dubbed as Dual Swap Disentangling (DSD), for disentangling that combines the best of the two worlds.” This is a questionable statement, as (already mentioned above) unsupervised disentangling has a different goal than supervised or semi-supervised one. Again, unsupervised disentangling is about discovery of latent factors, even -- and especially -- if they are not known to the designer of the network. This approach cannot discover any attributes that are not provided by ground truth labels and pre-assigned to specific groups of latents. While the argument seems to be made that attribute labels for pairs of images are generally easy to obtain, it seems to me that the opposite is the case, as soon as we leave the realm of fully synthetic images for which the ground truth is known by design. In fact, this problem is already apparent in this work, in the way the labeled MNIST pairs have to be created via a trained generative model (here InfoGAN) to get access to such labels. Furthermore, for the supervised part to work correctly, the attribute values of matched pairs need to be exactly equal (such as exactly matched azimuth angle, exactly matched color value, etc.) -- a problem that does start to show up even in this work under the term of “disturbed pair”. Line 243: “In this condition, we can compare part of codes (length = 5) that correspond to digit identity with whole codes (length = 5) of InfoGAN and β-VAE and variable (length = 1) that correspond to digit identity.” -- I have not been able to understand this sentence, which seems to be key for understanding how the quantitative results have been obtained. Generally, the quantitative evaluation section will need to be written much more clearly before the results can be considered convincing to many readers.

Reviewer 3



This papers presents a model that elicits interpretable disentangled representations from weakly supervised data. It does so through a dual-autoencoder model fit for both unlabelled and labelled data, where labelled data assumes knowledge of which part of the representation should be the shared one. The core of the model is the operation of swapping the shared part of the representation, when data is labelled, and dual-swapping of randomly selected parts of the representation in the case of unlabelled data. Strengths - the method makes sense, and is a well-thought combination of existing approaches (swapping, dual mechanism and parts of representations as labels) - it presents a nice way to add possibly softer labels, e.g. which part of the representation should be shared, as opposed to stronger labels (without forcing semantics) Weaknesses - the presented results do not give me the confidence to say that this approach is better than any of the other due to a lot of ad-hoc decisions in the paper (e.g. digital identity part of the code vs full code evaluation, the evaluation itself, and the choice of the knn classifier) - the results in table 1 are quite unusual - there is a big gap between the standard autoencoders and the variational methods which makes me ask whether there’s something particular about the classifier used (knn) that is a better fit for autoencoders. the particularities of the loss or the distribution used when training. why was k-nn used? a logical choice would be a more powerful method like svm or a multilayer perceptron. there is no explanation for this big gap - there is no visual comparison of what some of the baseline methods produce as disentangled representations so it’s impossibly to compare the quality of (dis) entanglement and the semantics behind each factor of variation - the degrees of freedom among features of the code seem binary in this case, therefore it is important which version of vae and beta-vae, as well as infogan are used, but the paper does not provide those details - the method presented can easily be applied on unlabeled data only, and that should have been one point of comparison to the other methods dealing with unlabeled data only - showing whether it works on par with baselines when no labels are used, but that wasn’t done. the only trace of that is in the figure 3, but that compares the dual and primary accuracy curves (for supervision of 0.0), but does not compare it with other methods - though parts of the paper are easy to understand, in whole it is difficult to get the necessary details of the model and the training procedure (without looking into the appendix which i admittedly did not do, but then, i want to understand the paper fully (without particular details like hyperparameters) from the main body of the paper). i think the paper would benefit from another writing iteration Questions: - are the features of the code binary? because i didn’t find it clear from the paper. if so, then the effects of varying the single code feature is essentially a binary choice, right? did the baseline (beta-)vae and infogan methods use the appropriate distribution in that case? i cannot find this in the paper - is there a concrete reason why you decided to apply the model only in a single pass for labelled data, because you could have applied the dual-swap on labelled data too - how is this method applied at test time - one needs to supply two inputs? which ones? does it work when one supplies the same input for both? - 57 - multi dimension attribute encoding, does this essentially mean that the projection code is a matrix, instead of a vector, and that’s it? - 60-62 - if the dimensions are not independent, then the disentanglement is not perfect - meaning there might be correlations between specific parts of the representation. did you measure/check for that? - can you explicitly say what labels for each dataset in 4.1 are - where are they coming from? from the dataset itself? from what i understand, that’s easy for the generated datasets (and generated parts of the dataset), but what about cas-peal-r1 and mugshot? - i find the explanation in 243-245 very unclear. could you please elaborate what this exactly means - why is it 5*3 (and not 5*2, e.g. in the case of beta-vae where there’s a mean and stdev in the code) - 247 - why was k-nn used, and not some other more elaborate classifier? what is the k, what is the distance metric used? - algorithm 1, ascending the gradient estimate? what is the training algorithm used? isn’t this employing a version of gradient descent (minimising loss)? - what is the effect of the balance parameter? it is a seemingly important parameter, but there are no results showing a sweep of that parameter, just a choice between 0.5 and 1 (and why does 0.5 work better)? - did you try beta-vae with significantly higher values of beta (a sweep between 10 and 150 would do)? Other: - the notation (dash, double dash, dot, double dot over inputs is a bit unfortunate because it’s difficult to follow) - algorithm 1 is a bit too big to follow clearly, consider revising, and this is one of the points where it’s difficult to follow the notation clearly - figure 1, primary-stage I assume that f_\phi is as a matter of a fact two f_\phis with shared parameters. please split it otherwise the reader can think that f_\phi accepts 4 inputs (and in the dual-stage it accepts only 2) - figure 3 a and b are very messy / poorly designed - it is impossible to discern different lines in a because there’s too many of them, plus it’s difficult to compare values among the ones which are visible (both in a and b). log scale might be a better choice. as for the overfitting in a, from the figure, printed version, i just cannot see that overfitting - 66 - is shared - 82-83 Also, require limited weaker…sentence not clear/gramatically correct - 113 - one domain entity - 127 - is shared - 190 - conduct disentangled encodings? strange word choice - why do all citations in parentheses have a blank space between the opening parentheses and the name of the author? - 226 - despite the qualities of hybrid images are not exceptional - sentence not clear/correct - 268 - is that fig 2b or 2a? - please provide an informative caption in figure 2 (what each letter stands for) UPDATE: I’ve read the author rebuttal, as well as all the reviews again, and in light of a good reply, I’m increasing my score. In short, I think the results look promising on dSprites and that my questions were well answered. I still feel i) the lack of clarity is apparent in the variable length issue as all reviewers pointed to that, and that the experiments cannot give a fair comparison to other methods, given that the said vector is not disentangled itself and could account for a higher accuracy, and iii) that the paper doesn’t compare all the algorithms in an unsupervised setting (SR=0.0) where I wouldn’t necessarily expect better performance than the other models.